# Dynamic Compressive Properties and Failure Mechanism of the Laser Powder Bed Fusion of Submicro-LaB6 Reinforced Ti-Based Composites

**DOI:** 10.3390/mi14122237

**Published:** 2023-12-13

**Authors:** Xianghui Li, Yang Liu

**Affiliations:** 1Faculty of Mechanical Engineering & Mechanics, Ningbo University, Ningbo 315211, China; 2Chaozhou Branch of Chemistry and Chemical Engineering Guangdong Laboratory, Chaozhou 515600, China; 3MOE Key Laboratory of Impact and Safety Engineering, Ningbo University, Ningbo 315211, China

**Keywords:** laser powder bed fusion, Ti-based composites, dynamic compressive properties, high strain rate, failure mechanism

## Abstract

In this study, lanthanum hexaboride (LaB_6_) particle-reinforced titanium matrix composites (PRTMCs, TC4/LaB_6_) were successfully manufactured using the laser powder bed fusion (LPBF) process. Thereafter, the effect of the mass fraction of LaB_6_ on the microstructure and the dynamic compressive properties was investigated. The results show that the addition of LaB_6_ leads to significant grain refinement. Moreover, the general trend of grain size reveals a concave bend as the fraction increases from 0.2% to 1.0%. Furthermore, the texture intensity of prior β grains and α grains was found to be weakened in the composites. It was also observed that the TC4/LaB_6_ have higher quasi-static and dynamic compressive strengths but lower fracture strain when compared with the as-built TC4. The sample with 0.5 wt.% LaB_6_ was found to have the best strength–toughness synergy among the three groups of composites due to having the smallest grain size. Furthermore, the fracture mode of TC4/LaB_6_ was found to change from the fracture under the combined action of brittle and ductility to the cleavage fracture. This study was able to provide a theoretical basis for an in-depth understanding of the compressive properties of additive manufacturing of PRTMCs under high-speed loading conditions.

## 1. Introduction

Laser powder bed fusion, as one of the most important additive manufacturing (AM) technologies, is able to build digitally designed parts up layer by layer by melting a feedstock using a micro-scale laser beam with a high energy density [1,2]. This unique production method provides more design freedom and flexible manufacturability compared with conventional fabrication methods [3,4]. Titanium alloys, in particular Ti-6Al-4V (TC4), have drawn enormous attention due to their high strength-to-weight ratio and excellent corrosion resistance [5]. Nowadays, the LPBF of TC4 is one of the most important research topics in AM, and it is mainly used in the fabrication of value-added components in the fields of aerospace, weapons, marine engineering, etc. [6,7]. However, the LPBF of TC4 suffers from the formation of hierarchical structures of acicular martensitic *α*′ grains within a large columnar of prior *β* grains, which results in inferior mechanical properties, especially under high-strain-rate conditions [8,9].

Particle-reinforced metal matrix composites (PRMMCs) are regarded as one of the alternatives to overcoming the performance defects of metals under impacting loads. However, conventional fabrication methods for PRMMCs—including powder metallurgy [10] mechanical alloying [11], self-propagation high-temperature synthesis [12], etc.—often have various shortcomings, such as interface cracks, segregation, etc. Recently, the continuous development of additively manufactured particle-reinforced Ti-based composite to foster their industrial adoption relies on achieving properties superior to the counterparts fabricated using conventional methods [13,14,15,16,17]. Rare earth (RE) or RE compounds are regarded as effective grain refiners, typically acting as heterogeneous nucleation sites for the *β* and *α* phases [14]. They induce the formation of new grains during the solidification process in additive manufacturing, leading to refined grain structures and impeding excessive grain growth. Barriobero et al. [15] found that the addition of La into Ti can change the texture orientation relationship between the *α* and parent *β* grains, resulting in the generation of equiaxed grains and weakening the anisotropy of LPBF titanium alloy. Bermingham et al. [16,17] found that the addition of LaB_6_ nanoparticles into LENS of TC4 reversed the direction of the Marangoni flow in the molten pool and refined the grain size, and the tensile strength was enhanced at a cost of ductility. Liu et al. [14] found that the addition of trace LaB_6_ into electron beam melting TC4 can reduce the size of prior *β* columnar and *α* grains as well as the texture intensity, thus reducing the anisotropy.

Dynamic impact resistance is one of the most important performance benchmarks of PRMMCs due to its wide application in the fields of aerospace, defense, etc. However, the literature shows that the dynamic mechanical properties of PRMMCs are closely related to the reinforcement and the interface between the matrix and the particles, which are different from the corresponding matrix [18]. Research on the dynamic mechanical response of AM titanium alloys had been reported frequently, encompassing aspects such as strength [19], toughness [20], constitutive behavior [21], dynamic damage [22], etc. However, there is still a notable gap regarding the investigation of the dynamic mechanical response of particle-reinforced titanium matrix composites in additive manufacturing. Therefore, Ti-based composites reinforced using the sub-micro particles were fabricated with LPBF through the mixing of TC4 with various mass fractions of LaB_6_. Furthermore, dynamic deformation behaviors were tested using the split Hopkinson pressure bar (SHPB) at various strain rates, and then the effects of the addition of LaB_6_ on the microstructure and dynamic compressive properties were revealed. These findings contribute to the understanding of the performances and deformation mechanism of LPBF of TC4/LaB_6_, which also provides a reference for designing high-performance particle-reinforced titanium matrix composites.

## 2. Experimental Details

### 2.1. Raw Materials

Gas-atomized spherical TC4 powder with an average particle size of 38.5 μm was used as the matrix, and irregular LaB_6_ particles (typically large particle sizes: 10–50 μm, average size: approximately 1 μm) were used as the reinforcement, as shown in Figure 1a,b. The two materials were mixed using planetary high-energy ball milling (NanDa Instrument QM-3SP4, Nanjing Nanda Instrument Co., Ltd., Nanjing, China). The mass fractions of the LaB_6_ particle were set as 0.2 wt.%, 0.5 wt.%, and 1.0 wt.%, respectively. The parameters of ball milling were set at a rotation speed of 350 rpm, milling time of 4 h, and ball-to-powder ratio of 4:1. From Figure 1c, it is observed that the composite powder maintains good sphericity. Additionally, Figure 1d shows that the nanoparticles adhere tightly to the surface of the Ti particles.

### 2.2. Specimen Preparation

The LPBF equipment used for the specimen preparation is the Concept Laser Mlab 100R, which consists of a 100 W continuous fiber laser (50 μm spot diameter), a high-accuracy galvo scanning system, a powder delivery system, a protective atmosphere filling device, and a circulation purification device. The forming chamber was filled with argon as a protective gas in order to ensure that the oxygen content in the chamber was below 100 ppm, preventing the metal material from oxidizing during the process. After a series of parameter evaluation experiments, the optimal process parameters were determined using the orthogonal scanning method, with a laser power of 72.5 W, scanning speed of 700 mm/s, powder layer thickness of 30 μm, and scanning hatch spacing of 70 μm. Furthermore, blocks with 5 × 5 × 5 mm^3^ were fabricated for the microstructure characterization, and a cylinder with *ϕ*5 × 5 mm^3^ was fabricated for the loading test. As shown in Table 1, the specimens were named TMC0, TMC1, TMC2, and TMC3 in accordance with the mass fractions of LaB_6_: 0 wt.%, 0.2 wt.%, 0.5 wt.%, and 1.0 wt.%, respectively.

### 2.3. Characterization and Analysis

Scanning electron microscopy (SEM, Hitachi SU5000, Hitachi High-Tech Co., Ltd, Tokyo, Japan), electron backscatter diffraction (EBSD, AZTEC 2.0, Oxford Instruments, Tubney Woods, Abingdon, Oxon OX13 5QX, UK), bright-field transmission electron microscopy (TEM, FEI Talos F200X, Beam Energy: 200 kV, Thermo Fisher Scientific, Waltham, MA, USA), and selected area electron diffraction (SAED, Selected Area Aperture: 10 μm) were used in this study in order to systematically characterize the microstructure, phase, and texture of the samples. Kroll reagent was used to etch the specimens. Electrolytic polishing was performed using Perchloric acid and Glacial acetic acid with a ratio of 1:19 at a voltage of 55 V and a current of 1.95 A. Ar ion beam thinning was used in the preparation of samples for TEM. A Materials Testing System (MTS 810) was used to perform the quasi-static compression tests at an initial strain rate of 10^−3^ s^−1^. A split Hopkinson pressure bar (SHPB) device was used to conduct impacting tests, the methods and test principles are specifically described in Ref. [20]. The impacting velocity was set as 15, 20, and 23 m/s, thus resulting in strain rates of about 1800, 2500, and 3000/s. The mechanical testing was conducted along the direction of sample fabrication, with each test condition repeated three times. Thereafter, the average value was taken into consideration.

## 3. Results and Discussions

### 3.1. Initial Microstructure

As shown in Figure 2a, it can be observed that the TMC0 exhibits distinct prior *β* columnar grain and grain boundaries (marked by white dashed lines in the figure), with a length of several millimeters and a width approximately equal to the hatch spacing. The formation of columnar grains is caused by epitaxial growth during the LPBF process [4,23]. A fine interlaced acicular *α*′/*α* phase with an average aspect ratio of 15–25 was formed within these prior *β* grains, and the grain size and orientation were found to be highly heterogeneous, which is in agreement with the findings of Refs. [14,17]. Although the traces of grain boundaries are still visible, the prior *β* grain of the TMC1 was found to be significantly narrowed, and a small number of pores formed in the grain. Moreover, the aspect ratios of acicular *α*′/*α* do not vary considerably (in the range between 15 and 25), and the direction is still strongly aligned (as shown in Figure 2b). When LaB_6_ increases to 0.5 wt.%, the prior *β* grain boundaries are no longer discernible in the TMC2, and a small number of acicular *α*′/*α* with decreasing aspect ratios are chaotically distributed, as shown in Figure 2c. When LaB_6_ further increases to 1.0 wt.%, most of the acicular *α*′/*α* disappear. This is consistent with Ref. [14], which revealed that an increase in LaB_6_ considerably decreased the width of the prior *β* grains as well as the aspect ratios of the acicular *α*′/*α* in the EBM of TC4 alloy. Marangoni convection is the dominant fluid flow that impacts the molten pool [24]. Due to steep temperature gradients in the molten pool, surface tension gradients are initially generated, leading to surface flow from regions of a lower surface tension to those of a higher surface tension [25]. Bermingham et al. [17] demonstrated that the addition of the RE element La can reverse the surface tension temperature coefficient, leading to a reversal of fluid flow direction, thereby influencing the molten pool. This impeded the epitaxial growth process of prior *β* columnar grains, ultimately leading to a more uniform microstructure.

In order to further identify the phase composition of LPBF TC4 and TC4/LaB_6_ composites, the specimens TMC0 and TMC3 were analyzed using bright-field transmission electron microscopy (TEM) and selected area electron diffraction (SAED). In Figure 3a, it is observed that the TMC0 is mainly composed of large grains with well-defined grain boundaries. The grains and boundaries in Figure 3a were analyzed using SAED, and the results show that the grains consist of *α*-Ti and the boundaries consist of *β*-Ti. This phenomenon is consistent with the findings of Ref. [26]. In order to determine whether the introduced reinforcing particle LaB_6_ can react with the Ti matrix, the TMC3 sample was analyzed using TEM and SAED, as shown in Figure 3d–f. Many nano-sized particles were formed in the *β*-Ti matrix (Figure 3d). The TEM image of the nano-sized particle at a high magnification (Figure 3e) shows that these particles are well combined in the matrix with clean interfaces and without debonding or cracking. Thereafter, the composition of nano-sized particles was determined to be La_2_O_3_ and *β*-Ti using SAED, as illustrated in Figure 3f. This indicated that LaB_6_ reacts with the matrix to produce the RE oxide La_2_O_3_. La_2_O_3_ serves as a heterogeneous nucleation site for the *β* phase, promoting the nucleation of the *β* phase and weakening the grain boundaries of the prior *β* columnar grains [27].

High-resolution transmission electron microscopy (HRTEM) was employed for an in-depth investigation of nanoparticles La_2_O_3_. As shown in Figure 4a, dislocations are observed at the boundaries between nanoparticles and the matrix. As shown in Figure 4b, multiple phases with different plane spacings be easily found in the magnified image of the nanoparticles’ boundaries. The lattice–fringe separations of 0.322 nm, 0.351 nm, and 0.366 nm were computed to identify the planes as (222) La_2_O_3_, (101) *β*-Ti, and (101) TiB, respectively. Regarding the parallel crystal planes of (222) La_2_O_3_ and (101) *β*-Ti, a mismatch occurs between the phase interfaces due to differences in interplanar distances. Consequently, dislocations are present in the phase interface, serving to ease the so-called mismatch [28,29].

The prior *β* columnar grains and grain boundary contours of the TMC0 are found to be more distinct on the EBSD inverse-pole figures (IPFs), as shown in Figure 5a. The fine acicular *α*′/*α* inside the prior *β* columnar grain are more noticeable. It was clear that most acicular *α*′/*α* could partially penetrate the *β* columnar grain boundary due to their length, demonstrating a well-organized internal structure and clean morphology. Because of the layer-by-layer manner of LPBF, the extremely high cooling rate resulted in distinct temperature difference between the adjacent prior *β* columnar grains. Then, it was observed that the *α* phase precipitated in the prior *β* grains as the temperature decreased, and the *α* grains were more likely to nucleate on the *β* grain boundaries, resulting in a lot of fine *α* equiaxed grain formations at the boundaries [30]. As shown in Figure 5b, the grain boundaries of the prior *β* columnar grain of the TMC1 are vaguely distinguishable, and the boundary areas of the distributed acicular *α*′/*α* start to show a certain degree of disorder. Moreover, many acicular *α*′/*α* also form in the intergranular regions of the prior *β* grains. When LaB_6_ increases to 0.5 wt.% and 1.0 wt.%, the prior *β* grains in the TMC2 and TMC3 are found to be indistinguishable, the acicular *α*′/*α* are significantly reduced and disordered, the number of *α* equiaxed grains rapidly increases, and they become randomly distributed, as illustrated in Figure 5c,d. As shown in Figure 5e–h, the four samples exhibited the highest texture intensity in the <0001> direction, according to the pole figures (PFs). Moreover, the TMC0 and TMC1 that could identify the prior *β* columnar grains have a higher texture intensity on the <0001> crystal planes than the remaining two samples that could not be used for this identification. The above results show that the addition of LaB_6_ weakens or even eliminates the grain boundaries of the prior *β* grains, refining the grains. This could be because the LaB_6_ particles could provide a large number of low-energy heterogeneous nucleation sites for *α* and *β* grains during the *L*→*β* and *β*→*α* + *β* phase transitions, resulting in a narrower *β* grain and eventually the formation of more equiaxed *α* grains [14,16].

Figure 5i exhibits the grain size obtained from the EBSD data, and the general trend of grain size comprises a concave bend when the mass fraction increases from 0 wt.% to 1.0 wt.%. The value of the TMC0 is 4.1 µm^2^, and the values decrease to 3.1 µm^2^ with the TMC1, and then to 2.0 µm^2^ with the TMC2. This indicates that a trace amount of LaB_6_ could cause a significant grain refinement on the LPBF TC4 alloy. However, the average grain size of the TMC3 was found to unexpectedly increase to 2.6 µm^2^. This trend is consistent with the findings of Refs. [31,32]. These studies found that, although the addition of LaB_6_ caused significant grain refinement in the TC4 alloy, the La_2_O_3_ particles tended to aggregate as the mass fraction exceeded a certain value, which had a negative influence on the grain refinement. This implies that there is an optimal mass fraction of LaB_6_ since the reinforcing particles range from 0.2 wt.% to 1.0 wt.%. Based on the results of this study, the mass fraction of 0.5 wt.% LaB_6_ is concluded as the optimal value for the LPBF of TC4/LaB_6_ composites.

### 3.2. Compression Mechanical Properties

#### 3.2.1. Quasi-Static Compressive Properties

The quasi-static stress–strain curves of LPBF TC4/LaB_6_ composites with different mass fractions are shown in Figure 6. In Figure 6a, it is observed that the specimens remain consistent within the elastic deformation stage, while a significant difference in the plastic deformation stage was observed. The yield strength of the samples is significantly enhanced due to the addition of LaB_6_. It was also observed that even a 0.2 wt.% addition could increase the yield strength from 857 to 954 MPa. However, when the mass fraction increases to 1.0 wt.%, the yield strength suddenly decreases. As mentioned before, LaB_6_ particles cause a remarkable grain refinement on the prior *β* columnar grains and the acicular *α*′/*α*. According to the Hall–Petch relationship, the yield strength increases as the grain size decreases, and this is called the refinement strengthening effect [33]. Moreover, the finer grains imply that there are more grain boundaries in the samples, and more grain boundaries effectively prevent the movement of dislocations and thereby improve the yield strength. The average grain size decreases from 4.1 to 2.0 µm^2^ in the TMC2 when compared to the TMC0, and the yield strength increases from 857 to 1312 MPa as a consequence. Thereafter, with the further addition of LaB_6_, the agglomeration of La_2_O_3_ is found to make the average grain size of the TMC3 increase to 2.6 µm^2^, resulting in a decrease in the yield strength. Figure 6b illustrates that the fracture strain decreases monotonously with the increasing LaB_6_. This is due to an increase in the mass fraction of LaB_6_ causing a greater distribution of the reinforced particles in the samples. Furthermore, the corresponding dispersion strengthening effect is more apparent, which makes the yield strength evenly increase and the grain size coarsen. Simultaneously, the compressive load tends to cause a concentration of stress on the interface between the particle and matrix. when the stress exceeds the yield limit, the generation, growth and connection of microcracks are bound to happen on the interface, which makes the sample more fragile and prone to brittle failure.

#### 3.2.2. Dynamic Compression Properties

The dynamic compressive stress–strain curves of the TC4 and TC4/LaB_6_ samples at strain rates of 1800/s, 2500/s, and 3000/s are shown in Figure 7. All curves have the same shape, and the fluctuation in the curves is caused by the repeated oscillations of stress waves in sample [34]. Moreover, it was found that the work hardening rates (UCS-YS) at high strain rates are significantly higher than those under quasi-static conditions. The excellent work hardening ability is advantageous in structural applications in order to guarantee a large safety margin before the fracture [35,36]. Although the dynamic YS are lower than those under quasi-static conditions, the ultimate compressive strength (UCS) is apparently higher than that of the former (compared with Figure 6), showing a significant strain rate sensitivity. Similar to the quasi-static condition, at the three strain rates, the dynamic compressive strength is found to initially increase as the mass fraction of LaB_6_ increases from 0 to 0.5 wt.%, and then drops as LaB_6_ further increases to 1.0 wt.%. From Figure 7b,d,f, it is observed that the change trends in dynamic YS, UCS, and fracture strain coincided well with the trend of the grain size. Unlike the quasi-static condition, the fracture strain of the TMC3 under the dynamic condition does not continue to decrease, but instead increases.

It is also found that the fracture strain decreased at high strain rates, compared with Figure 6. This is due to the plastic localization of titanium alloy under impact loading, thus deteriorating plastic deformation [26,37]. Furthermore, when the mass fraction of LaB_6_ is relatively higher, more micro pores are formed in the TC4/LaB_6_. These pores and cracks play an important role in dynamic compressive properties. Under high-speed impact loading, the release of the stress is caused due to the nucleation of micro cracks and increases fracture toughness [38], whereas, since the impact resistance of alloys is less sensitive to pores and microcracks than tensile properties [39], some pores that are close to each other may even collapse, merge, and close, thereby improving the fracture strain of the alloys under impact loading conditions.

#### 3.2.3. Influence of Grain Size on the UCS

The Hall–Petch laws define the relationship between grain size (*d*^−1/2^) and yield strength. In Figure 6 and Figure 7, it is observed that a positive relationship exists between the grain size and strength of LPBF of TC4 and TC4/LaB_6_ samples. However, there is no definite yield point in the stress–strain curves under dynamic conditions. Hence, the UCS is selected as the basis for comparison instead, as shown in Table 2 and Figure 8. Under the influence of strain rate strengthening [36], the dynamic UCS is higher than the quasi-static state, but the change trend is similar on the whole. With the decrease in grain size, UCS increases monotonously. Then, the grain size of the TMC3 increases due to La_2_O_3_ agglomeration, and UCS decreases accordingly. It is also indicated that UCS is closely related to grain size. As shown in Figure 5i, the addition of LaB_6_ effectively reduced the grain size. This is mainly attributed to the enrichment of La and its oxides at the grain boundaries, which hindered the growth of *β*-Ti, known as pinning effect. Moreover, they could serve as heterogeneous nucleation sites for the *α* phase and promote its nucleation and growth [31,32]. The effective slip distance of dislocation decreased, and the hindrance effect of grain boundary increased due to the grain refinement caused by the introduction of the reinforcing phase. The reinforced particles also hindered the dislocation, and thereby the energy required for the dislocation slip increased. Moreover, due to the limited slip systems of the *α* phase with an HCP structure, twinning is usually caused in places where stress is concentrated and when the slip is impeded under high-strain-rate deformation [40,41]. Yu et al. [42] suggested that the Hall–Petch slope k of deformation twinning is larger than that of the dislocation slip in titanium matrix composites. According to Hall’s dislocation pile-up model [43], dislocations piled up in the vicinity of the grain boundary, and the stress concentration from the dislocation pile-up produced a dislocation source in an adjacent grain [44]. Grain refinement resulted in the increase in grain boundaries of the *α* phase, so the activation energy required for dislocation propagation in the adjacent grains enhanced. Thus, the UCS of TC4/LaB_6_ effectively improved. This was consistent with the results of Wang et al. [45], who found that the strength of titanium matrix composites can be significantly improved after matrix grain refinement.

### 3.3. Fracture Characteristics

The SEM images of the fracture surfaces of the TMC0 and TMC2 are exhibited in Figure 9. A staggered distribution of dimples and smooth shear zones are observed in the TMC0 at low magnification (Figure 9a), which is a typical mixed fracture mode of ductile and brittle fracture [46]. The same phenomenon was reported in the EBM of TC4 alloy [47,48] because the *α*′ phase has a higher strength but lower ductility, thus showing brittleness [49]. At a higher magnification, it is found that the dimples had a elongated shapes and sizes of 2~20 µm (Figure 9b), which were caused by the shear stress in the shear direction [50]. As shown in Figure 9c, cleavage steps and planes are alternately distributed on the whole fracture surface of the TMC2, which is a typical cleavage fracture. At higher magnification (Figure 9d), small dimples with smaller sizes are observed near the step surface, most of which are found to be less than 1 µm and uniformly distributed due to the uniform distribution of LaB_6_ particles in the matrix. The size of the dimple is a critical indicator for assessing material ductility, reflecting the material’s energy absorption capability and toughness characteristics. Larger dimples indicate better energy absorption and enhanced ductility performance [26,36]. In this study, TC4/LaB_6_ exhibited smaller dimples compared to LPBF of TC4, indicating a lower ductility, which is consistent with the failure strain results shown in Figure 6b.

The 3D morphologies of the samples under an impact load at a strain rate of 2500/s are depicted in Figure 10, where blue arrows indicate convexity and black arrows indicate concavity. After the impact compression, all the samples are shear fractured with different fracture characteristics. The TMC0 was found to be incompletely fractured (Figure 10a; external force is needed to separate the two parts). Additionally, the TMC1 and TMC2 were fully fractured along 45° (the direction of maximum shear stress), and the samples fractured into two fragments (Figure 10b,c). The two fragments of the TMC1 have rough convex and concave surfaces, while those of the TMC2 have relatively flat fracture surfaces. Correspondingly, the TMC3 shear fractures into several fragments with different sizes, in which only the two largest pieces were selected after the fracture, as shown in Figure 10d. Some larger-size fracture plateaus and undulating surfaces on the fracture surface of the TMC3 were also found.

In order to characterize the fracture models of LPBF of TC4 and TC4/LaB_6_ under impact loading conditions, a scanning electron microscope (SEM) was used to further observe the fracture surfaces at a strain rate of 2500/s, as shown in Figure 11. It was observed that the morphology of the TMC0 is quite different from those of TC4/LaB_6_ composites. As shown in Figure 11a, large dimple areas and smooth areas are alternatively distributed on the fracture surface. Dimples with sizes of 2~10 μm (less than those at 10^−3^/s) are elongated in the shear direction, indicating that the TMC0 is mainly characterized by mixed ductile/brittle fracture. This kind of fracture model is also observed in some studies on the AM of a TC4 alloy [20].

Correspondingly, a large smooth dimple area is no longer found in the TC4/LaB_6_ samples; instead, large-sized cleavage steps and cleavage planes are visible, and the surfaces are no longer flat and smooth. The higher-magnification SEM images show that dense featureless dimples are distributed on the cleavage planes. These dimples are so fine (less than 2 μm) that TC4/LaB_6_ cannot be characterized under ductile fracture. The dimple size is associated with the grain size because the grain boundaries function as crack nucleation sites that influence the size of the dimples [27,51]. When compared with the TCM0, TMC1 starts to show cleavage steps, and the dimples on it become dense and irregular (Figure 11b), indicating that the reinforcing particles hinder crack slippage during the dynamic loading conditions [10,52]. It is also observed that the TMC2 surface starts to show cleavage steps with a larger size, whereas higher-magnification images show small and deep featureless dimples (Figure 11c) and particulate matter (yellow arrows in Figure 11c) present on cleavage planes. This indicates that the hindering effect of the reinforcing particles on the crack slip during dynamic loading is further enhanced. The TMC3 surface exhibited the most cleavage fracture characteristics. The SEM image at high magnification shows the presence of rough dimples on the cleavage planes (Figure 11d), and a large number of particles filled inside (yellow arrows in Figure 11d). Too many particles reduced the mechanical properties of the TMC3, which coincides with the reduction in its UCS.

## 4. Conclusions

This paper exposes the impact of LaB_6_ content on the microstructure, quasi-static characteristics, and dynamic properties of titanium matrix composites, focusing primarily on the TC4 and TC4/LaB_6_ composites manufactured using LPBF. Based on the results, we made following conclusions:(1)A small amount of LaB_6_ had a considerable effect on the microstructure of the titanium matrix composites, and 0.2 wt.% LaB_6_ weakened and blurred the prior *β*-columnar grain boundaries and acicular *α*′/*α*. LaB_6_ reached a critical point at 0.5 wt.%, where the boundaries of the prior *β*-columnar grains became indistinguishable, and the acicular *α*′/*α* decreased rapidly. When it increased to 1.0 wt.%, the influence on the change in microstructure was significantly diminished. The addition of LaB_6_ had a significant grain refinement effect, but when its mass fraction exceeded 0.5 wt.%, the nano-reinforced particulate tended to agglomerate, which had a negative impact on grain refinement.(2)Benefiting from the fine-grain strengthening and dispersion strengthening of LaB_6_, the quasi-static compressive strength of the TC4/LaB_6_ was found to be significantly higher than that of the pure TC4. However, it also led to a decrease in plasticity, especially when the LaB_6_ fraction was relatively high (1.0 wt.%), and the specimens fractured, even shortly after the end of the elastic segment.(3)The performance of TC4/LaB_6_ was related to the mass fraction of LaB_6_, but higher did not mean better. A critical mass fraction of 0.5 wt.% was found in this study. Excessive LaB_6_ significantly increased the brittle fracture characteristics of the material, showing a more significant terrace-like fracture surface after extrusion and friction.

## Figures and Tables

**Figure 1 micromachines-14-02237-f001:**
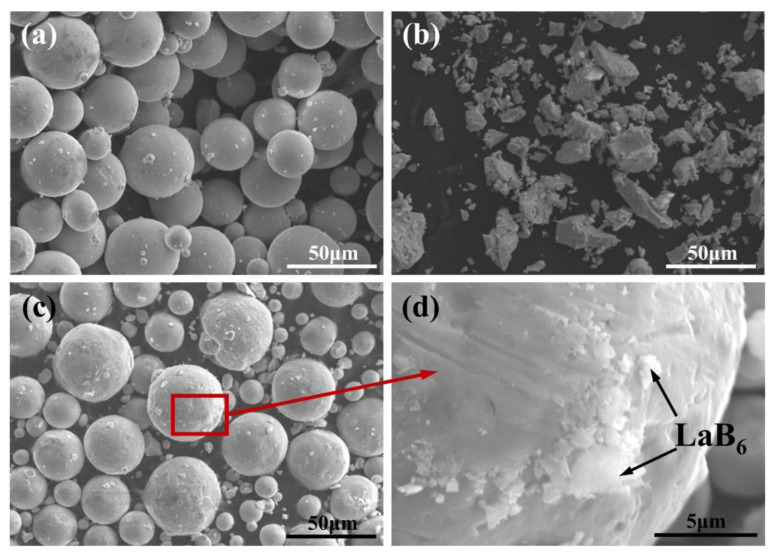
SEM images of raw powders: (**a**) gas-atomized spherical TC4 powder, (**b**) irregular LaB_6_ nanoparticle, (**c**) composite powder after ball milling, and (**d**) higher magnification image showing the nanoparticles adhering to the surface of Ti particles.

**Figure 2 micromachines-14-02237-f002:**
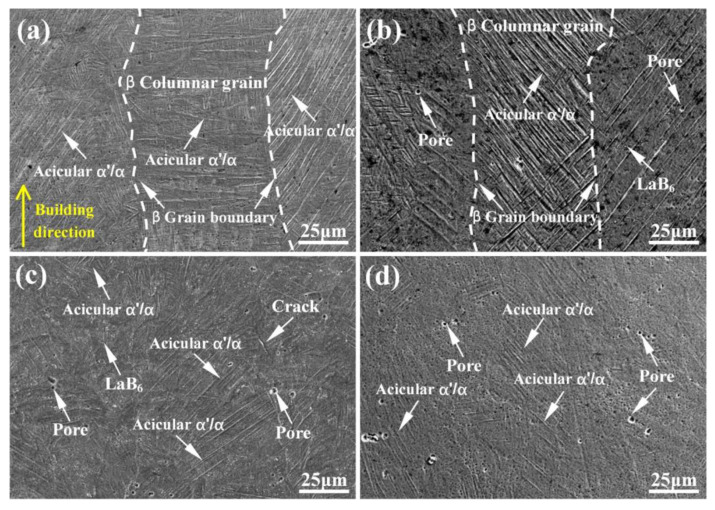
SEM images of microstructure of (**a**) TMC0, (**b**) TMC1, (**c**) TMC2, and (**d**) TMC3.

**Figure 3 micromachines-14-02237-f003:**
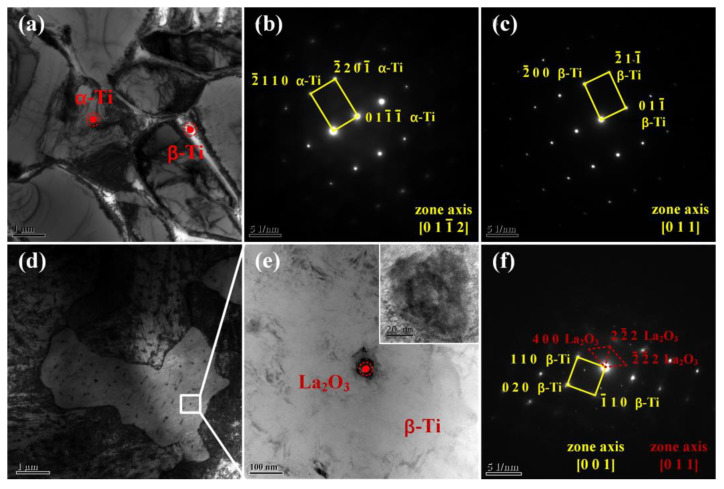
Bright-field TEM images of LPBF samples and corresponding selected area electron diffraction patterns: (**a**–**c**) TMC0, (**d**–**f**) TMC3.

**Figure 4 micromachines-14-02237-f004:**
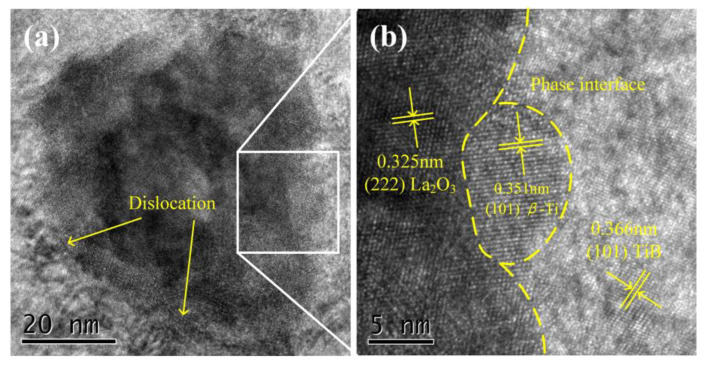
(**a**,**b**) High-resolution transmission electron microscopy (HRTEM) images of the nano-sized particle of TMC3.

**Figure 5 micromachines-14-02237-f005:**
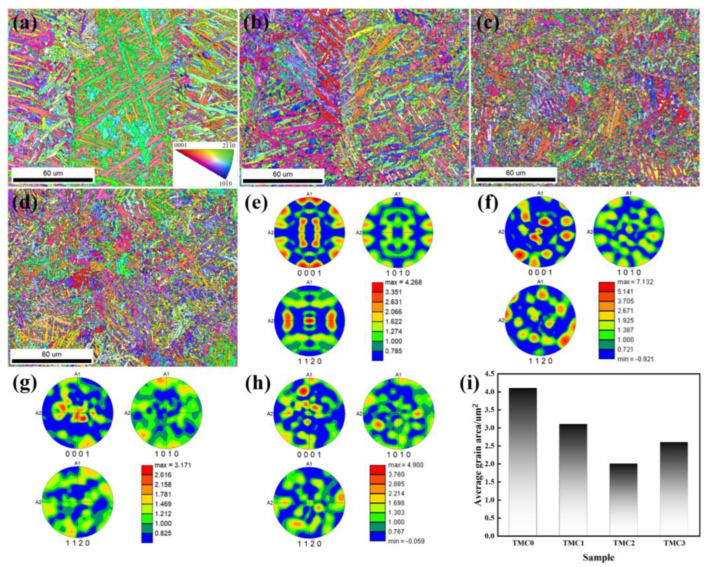
EBSD inverse pole figures (IPF), pole figures (PF), and the average grain size of LPBF samples: (**a**,**e**) TMC0, (**b**,**f**) TMC1, (**c**,**g**) TMC2, (**d**,**h**) TMC3, and (**i**) average grain size.

**Figure 6 micromachines-14-02237-f006:**
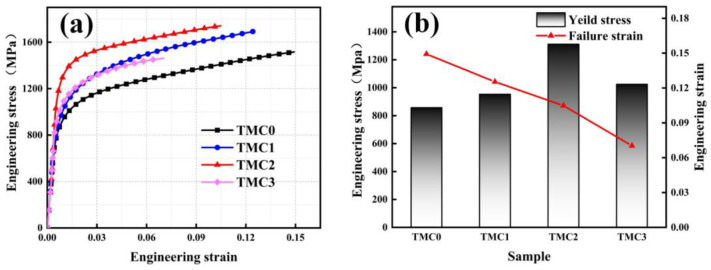
Quasi-static compressive properties of TC4/LaB6 with different mass fractions: (**a**) engineering stress–strain curves; (**b**) compressive 0.2% proof stress and failure strain.

**Figure 7 micromachines-14-02237-f007:**
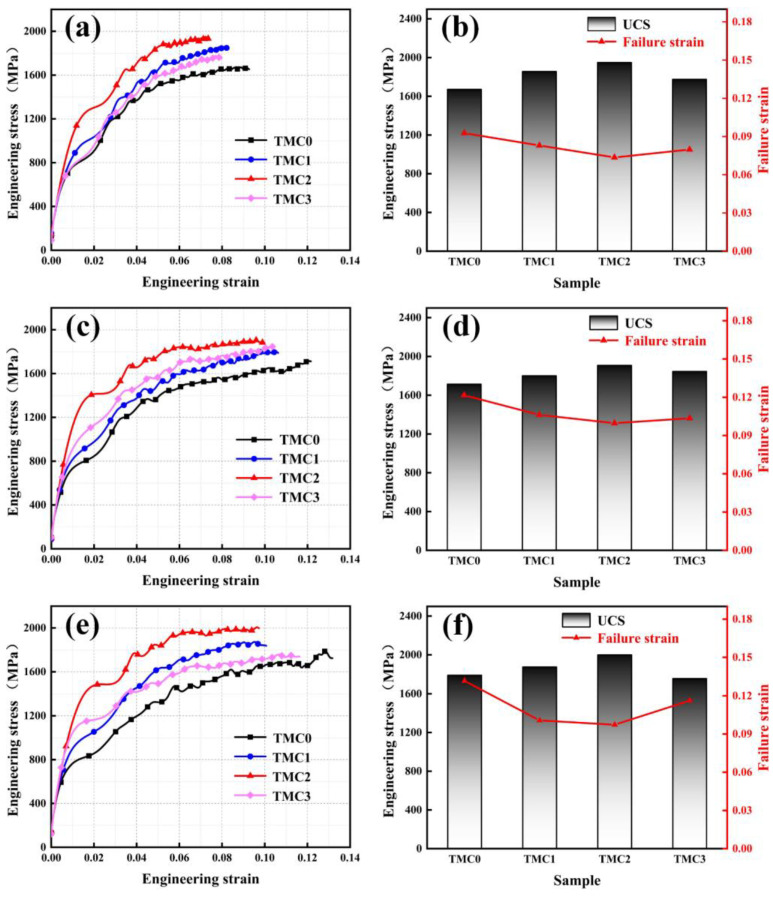
Dynamic compressive properties of LPBF of TC4/LaB6 composites with different mass fractions at different strain rates (1800/s, 2500/s and 3000/s): (**a**,**c**,**e**) engineering stress–strain curves; (**b**,**d**,**f**) ultimate compressive strength and failure strain.

**Figure 8 micromachines-14-02237-f008:**
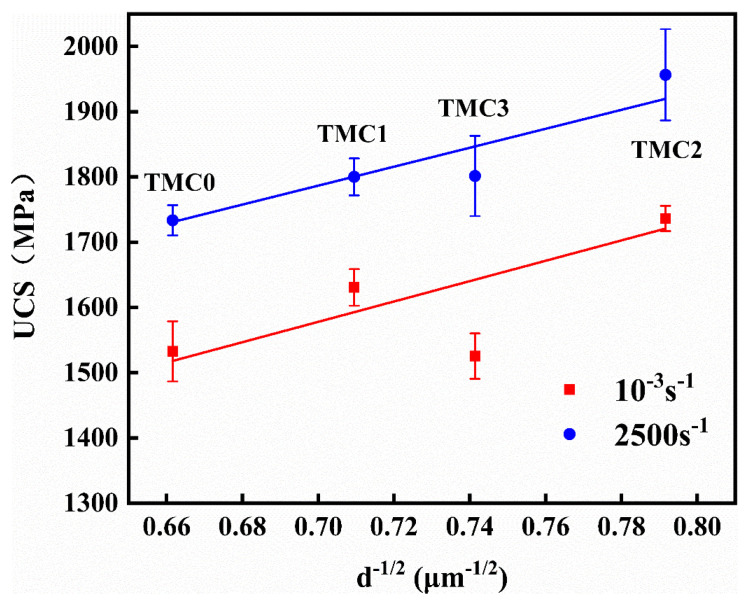
Ultimate compressive strength for strain rates of 10^−3^/s and 2500/s.

**Figure 9 micromachines-14-02237-f009:**
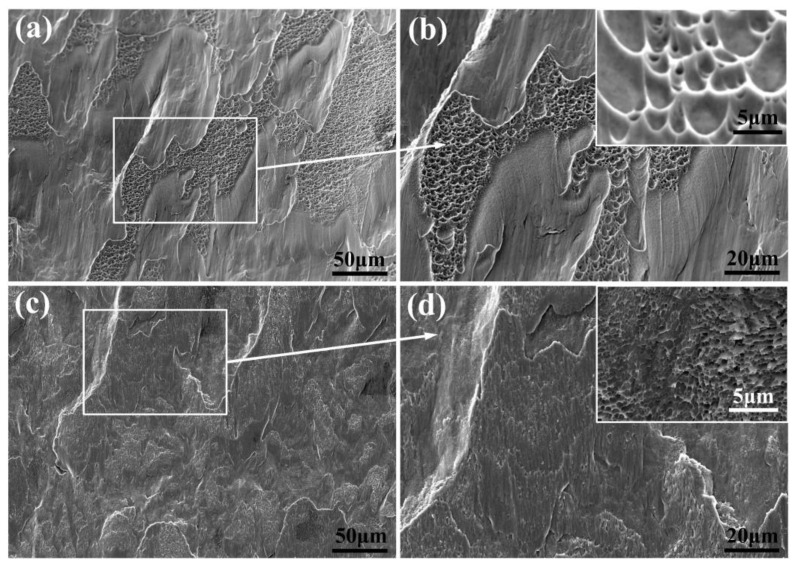
The fracture surfaces of compressed samples of TMC0 and TMC2: (**a**,**b**) the fracture surface of TMC0; (**c**,**d**) the fracture surface of TMC2. The inset showed detailed dimple morphologies at higher magnifications.

**Figure 10 micromachines-14-02237-f010:**
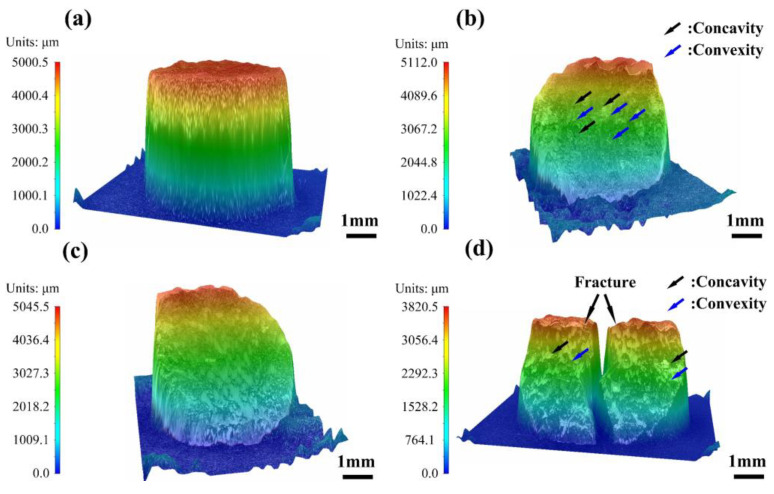
Thre-dimensional topographies of samples with different mass fractions after shock compression at a strain rate of 2500/s: (**a**) TMC0, (**b**) TMC1, (**c**) TMC2, and (**d**) TMC3.

**Figure 11 micromachines-14-02237-f011:**
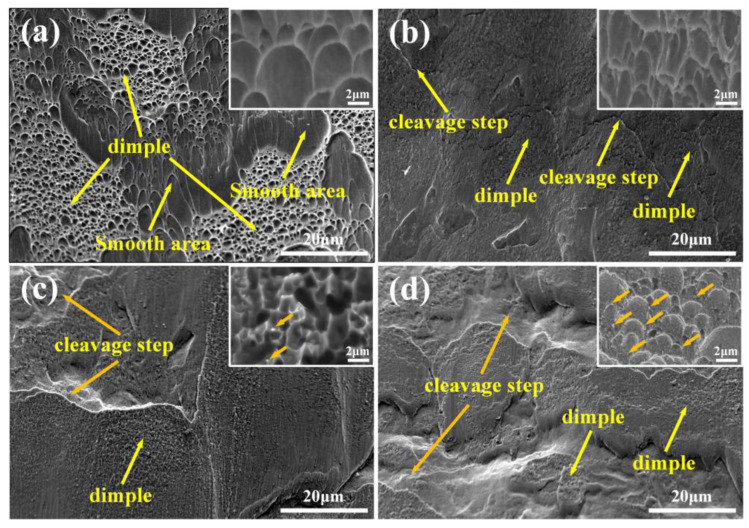
SEM images of samples with different mass fractions after an impact load at a strain rate of 2500/s: (**a**) TMC0, (**b**) TMC1, (**c**) TMC2, and (**d**) TMC3. The inset shows detailed dimple morphologies at higher magnifications.

**Table 1 micromachines-14-02237-t001:** Different names corresponding to various LaB_6_ mass fractions.

Name	Mass Fraction (wt.%)
TMC0	0
TMC1	0.2
TMC2	0.5
TMC3	1.0

**Table 2 micromachines-14-02237-t002:** Ultimate compressive strength and grain size for strain rates of 10^−3^/s and 2500/s.

Name	UCS (MPa)		Grain Size *d*^−1/2^ (μm^−1/2^)
	Quasi-Static (10^−3^/s)	Dynamic (2500/s)	
TMC0	1532.7 ± 32.7	1733.6 ± 16.4	4.1
TMC1	1630.6 ± 19.8	1800.1 ± 20.1	3.1
TMC2	1736.3 ± 13.6	1956.3 ± 49.5	2
TMC3	1525.4 ± 24.5	1801.7 ± 43.4	2.6

## Data Availability

Data are contained within the article.

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
