# Peer review of "Dynamic Compressive Properties and Failure Mechanism of the Laser Powder Bed Fusion of Submicro-LaB6 Reinforced Ti-Based Composites"

_micromachines, 2023, doi:10.3390/mi14122237_

Round 1
Reviewer 1 Report
Comments and Suggestions for Authors
This paper mainly investigated the effects of LaB6 inoculant addition on the microstructure, compressive properties and impact properties of the LPBF-fabricated TC4 alloy sample. This work presented a systematic investigation and showed to some extent innovation. The following points need to be considered:
1. There existed some incorrect statements, such as “there are few reports concerning the AM of reinforced Ti-based composite” (line 66), “equiaxed prior βgrains” (line 125).
2. In experimental details section, the LaB6 particle size seems to be more than 1 μm from Fig. 1b, apparently not belonging to a nano scale which is usually smaller than 100 nm.
3. The authors stated that the presence of RE elements resulted in a changed molten pool shape. However, it seemed to be difficult to observe this change from Fig. 2.
4. Whether did the authors observe the rare earth oxide at the grain boundaries of the priorβphase or not? The corresponding evidence should be provided. Besides, the authors are suggested to provide the HRTEM image of the interface between the matrix and nanoparticle to disclose the lattice misfit.
5. In views of a significant increase in the maximum texture index from 4.268 to 7.132, it is difficult to be concluded that the addition of LaB6 reduces the anisotropy of LPBF TC4.
6. There is still large room for improvement in the writing style and English language.
Author Response
Dear editors and reviewers,
The authors would like to extend their extreme gratitude for your kind comments and constructive feedback on our manuscript entitled “Dynamic compressive properties and failure mechanism of laser powder bed fusion of submicro-LaB6 reinforced Ti-based composites”.
The authors accept most of the views on the original paper given by the reviewers and editor. Your comments have been considered carefully and we have made some modifications to the manuscript accordingly. The corresponding revisions within the body of the manuscript were highlighted in yellow.
Thank you very much and best regards.
Yours sincerely,
Yang Liu, liuyang1@nbu.edu.cn
Please refer to the attachment for the detail responses your comments.

Reviewer 2 Report
Comments and Suggestions for Authors
This paper investigated the effect of the mass fraction of LaB6 on the microstructure and the dynamic compressive properties of LPBF Ti-base composites. This work is interesting, however, there are several points that needs to be addressed.
1. In section 2.3 there is no information about the TEM analysis, what microscope make and model was used, the accelerating voltage, how the samples were prepared, no how big a selected area aperture was used.
2. In figure 2 why the melt pool boundaries are not visible in c) and d) and what is the growth direction?
3. In figure 2 - how were the arrowed "LaB6" regions identified? Was this by EDX or by EBSD or is it speculation?
4. In figure 3 the diffraction spots are vectors and should be shown by convention without brackets which are reserved for planes when using Miller indices.
5. In Figure 3 - why no examination of TMC2 if this was the best performing sample?
6. Page 8 line 21: "inner area": is meant to mean the area inside or in between the grains?
7. Page 8 line 25: this sentence doesn't make sense. Does "rapidly increased" refer to the size? If so it needs to be clear.
8. Page number in the some references appear incomplete.
Comments on the Quality of English Language
Minor editing of English language required.
Author Response

(The authors gave the same response as above.)

Reviewer 3 Report
Comments and Suggestions for Authors
The paper discusses the dynamic compression properties and failure mechanism of laser powder bed fusion of titanium matrix composites reinforced with lanthanum hexaboride (LaB6) particles. This study is characterized because the authors studied the deformation mechanism by LPBF of TC4/LaB6. In this research, dynamic deformation behaviours, microstructural analysis and dynamic compression properties were studied. The article is interesting for the method of obtaining it, and provides concepts of dynamic compression properties and failure mechanism of the alloy (PRTMCs, TC4/LaB6), however, the article has several details and errors that must be corrected, and some questions need to be answered before publication.
Page 3. Line 97. What was the energy density and wavelength used? You should explain why do you use a continuous fiber laser, and not another type of laser such as NdYAG or CO2?.
Page 2. Line 75. Although figure 1 has a lot of similarity (i.e. both figures are different but have similarity) with figure 1 of "Microstructure and Mechanical Properties of LaB6/Ti-6Al-4V Composites Fabricated by Selective Laser Melting", This article should have been cited, at least because the topic is of interest. Likewise, the authors must explain what the difference is with this work.
Page 4. Line 146. Although the authors explained why there was a change in the microstructure of the phases (acircular α′/α, and grains, β columnar 120 grain), which is attributed to the formation of some cracks, including an increase in porosities with 1% LaB6 (Fig2 (d)).
Page 6. Line 201-202. To which is attributed the grain refinement on the LPBF TC4 alloy, when LaB6 is added up to 0.5% wt.?
When the previous points are answered or attended the article may be published.
Author Response

(The authors gave the same response as above.)
